

# Quantitative trait loci associated with straighthead-resistance used for marker assisted selection in rice (*Oryza sativa* L.) RIL populations

Xuhao Pan[1,2], Yiting Li[1] and Xiaobai Li[3]

[1] Tobacco Research Insistitute of Chinese Academy of Agriculture Sciences, Qingdao, China
[2] Rice Institute, Sichuan Agriculture University, Chengdu, China
[3] Zhejiang Academy of Agricultural Sciences, Zhejiang, China

## ABSTRACT

Straighthead is a physiological disorder of rice (*Oryza sativa* L.) that causes dramatic yield loss in susceptible cultivars. This disorder is found worldwide and is reported to increasingly occur in the southern United States. Genetic resistance breeding has been considered as one of the most efficient methods for straighthead prevention because the traditional prevention method wastes water and costs labor. In this study, we analyzed the genetic effects of five straighthead quantitative trait loci (QTLs), namely, AP3858-1 (*qSH-8*), RM225 (*qSH-6*), RM2 (*qSH-7*), RM206 (*qSH-11*), and RM282 (*qSH-3*), on the recombinant inbred lines (RILs) developed from Jing185/Cocodrie and Zhe733/R312 populations using our five previously identified markers linked to these QTLs. As a result, recombinant inbred lines (RILs) with four resistant alleles at the four loci (AP3858-1, RM225, RM2, and RM206) exhibited the highest straighthead resistance. This result suggests that the four markers could be efficiently used to select the straighthead-resistant recombinant inbred lines (RILs). Furthermore, by using AP3858-1, we successfully obtained five straighthead-resistant recombinant inbred lines (RILs) with more than 50% genetic similarity to Cocodrie. These markers and recombinant inbred lines (RILs) can be used for future straighthead resistance breeding through marker-assisted selection.

## INTRODUCTION

Straighthead is a physiological disorder of rice that is characterized by sterile florets and distorted spikelets (*Yan et al., 2005*). It can make rice kernels empty and panicles erect and fail to head out. As a result, straighthead often causes dramatic yield loss in susceptible cultivars (*Dilday et al., 2000*). Straighthead was first reported in the US (*Wells & Gilmour, 1977*) and is now found in Japan (*Takeoka, Tsutsui & Matsuo, 1990*), Australia (*Dunn et al., 2006*), Portugal (*Cunha & Baptista, 1958*), Thailand (*Weerapat, 1979*), and Argentina (*Yan et al., 2010*). It has become a huge threat to rice production in the southern US and worldwide.

Corresponding author
Xuhao Pan, panxuhao@caas.cn

According to previous studies, straighthead can be caused by numerous factors, such as sandy to silt loam-textured soils (*Ehasanullah & Meetu, 2018*), low free iron and low pH in soil (*Hua et al., 2011*; *Huang et al., 1997*), presence of As, Mn, Ca, and S, and soil organic matter (*Hua et al., 2011*; *Hulbert & Bennetzen, 1991*). In the southern U.S., arsenic-based herbicides such as monosodium methanearsonate (MSMA) have been widely applied in cotton-growing areas. Thus, arsenic (As) usually residues in paddies. Toxicity in rice induces a series of symptoms, such as decreases in plant height and tillers (*Kang et al., 1996*), reduction in shoot and root growth (*Dasgupta et al., 2004*; *Rahman et al., 2012*), inhibition of seed germination (*Shri et al., 2009*; *Rahman et al., 2012*), decline in chlorophyll content and photosynthesis, and sometimes plant death (*Rahman et al., 2007*). Notably, As can cause typical straighthead symptoms in susceptible rice cultivars in MSMA-applied soil (*Rahman et al., 2008*; *Lomax et al., 2012*). Thus, MSMA-induced application is a common method of evaluating rice straighthead (*Slaton et al., 2000*; *Wilson Jr et al., 2001*).

For straighthead prevention, one method used is water management called "draining and drying" (D&D). In this method, farmers need to drain their rice field about two weeks after a permanent flood and then wait for reflooding until the rice leaves exhibit drought stress symptoms (*Rasamivelona, Kenneth & Robert, 1995*; *Slaton et al., 2000*). In Arkansas, one-third of the rice fields applies the D&D method, which results in approximately 150 million $m^3$ of wasted irrigation water every year (*Wilson Jr & Runsick, 2008*). Clearly, the method costs natural resources and manpower and also leads to drought-related yield loss.

Resistant breeding is considered as the most efficient and environmentally friendly strategy for straighthead prevention. A number of resistant germplasms have been identified, and the genetic base of straighthead has been examined (*Yan et al., 2002*; *Pan et al., 2012*). Marker-assisted selection (MAS) has been used in resistant breeding for many years and has been demonstrated as a feasible strategy in multiple crops (*Yan et al., 2005*). In our previous study (*Pan et al., 2012*), we constructed two recombinant inbred line (RIL) $F_9$ populations using two resistant parents (Zhe733 and Jing185) and the susceptible parents Cocodrie and R312. Five quantitative trait loci (QTLs), namely, *qSH-3*, *qSH-6*, *qSH-7*, *qSH-8*, and *qSH-11*, were identified to be associated with straighthead *via* linkage mapping using the two RIL populations. Four QTLs (*qSH-6*, *qSH-7*, *qSH-8*, and *qSH-11*) were determined for the Zhe733/R312 population and two QTLs (*qSH-3* and *qSH-8*) were identified for the Cocodrie/Jing185 population. Of these QTLs, *qSH-8*, which is 290 kb long and is found on chromosome 8, was identified in the two populations. Moreover, the presence of *qSH-8* was confirmed in the $F_2$ and $F_{2:3}$ populations of Zhe733/R312 (*Li et al., 2016b*). Therefore, *qSH-8* was proven as a major QTL for straighthead resistance. Furthermore, five markers, namely, RM282, RM225, RM2, AP3858-1, and RM206 (Table S1), were associated with the five aforementioned QTLs, respectively.

Arkansas accounts for a large part of rice production in the U.S.. However, as previously mentioned, many cultivars grown in this region are highly susceptible to straighthead. For instance, Cocodrie, a major cultivar grown in Arkansas, lost up to 94% of its yield when straighthead occurred (*Linscombe et al., 2000*; *Wilson Jr et al., 2001*). Thus, genetically improving straighthead resistance is necessary to ensure high rice yields. In the present

study, our objective is to identify RILs with straighthead-resistant QTLs and similar agronomic traits and backgrounds to Cocodrie in the Cocodrie/Jing185 population for use in further resistant breeding.

## MATERIALS & METHODS

### Plant material

Two RIL $F_9$ populations, Zhe733/R312 and Cocodrie/Jing185, were previously developed and evaluated for straighthead (Fig. 1) (*Pan et al., 2012*). Resistant cultivars Zhe733 (PI 629016) and Jing185 (PI 615205) and susceptible cultivar R312 (PI 614959) were from China. Cocodrie (PI 606331), another susceptible cultivar, is widely grown in the US. All three cultivars from China belong to *indica*, whereas Cocodrie belongs to *japonica*. A total of 170 $F_9$ RILs were identified in the Zhe733/R312 population, whereas 91 $F_9$ RILs were produced in the Cocodrie/Jing185 population.

### Phenotyping

Both Zhe733/R312 and Cocodrie/Jing185 populations were planted in MSMA-treated soil at Dale Bumpers National Rice Research Center near Stuttgart, Arkansas for two years (2010 and 2011). Using a randomized complete block design, the RILs of the two $F_9$ populations were planted in single-row field plots (0.62 m$^2$) with three replications, as previously described (*Pan et al., 2012*). Exactly 6.7 kg ha$^{-1}$ of MSMA was applied to the soil surface and incorporated prior to planting, as previously described (*Yan et al., 2005*). The four parents (Zhe733, R312, Cocodrie, and Jing185) were repeatedly planted in each field tier of 99 rows as controls. Field management was performed as previously described (*Yan et al., 2008*).

Evaluation of straighthead rating was based on floret sterility and panicle development using a scale of 1 to 9 at the maturity stage (*Yan et al., 2005*). A score of 1 represented normal plants with panicles fully emerged and more than 80% grains developed, whereas 9 represented sterile plants with no panicle emergence and complete absence of developed grains. Based on our previous research, RILs with a score of 4.0 or below were resistant and had 41%–60% seed sets or higher, whereas RILs with a score of 6.0 or above were susceptible and had 11%–20% seed sets or lower (*Li et al., 2016b*).

The Cocodrie/Jing185 population was then planted in clean soil without MSMA at Dale Bumpers National Rice Research Center near Stuttgart, Arkansas for two years (2010 and 2011). To ensure a reliable evaluation, we performed water management to prevent straighthead. We conducted a randomized complete block design for the field experiments. RILs were planted in single-row field plots (0.62 m$^2$) with three replications each year. The parents were repeatedly planted in a field tier of 99 rows as controls.

Evaluations of the heading date, height, and tillers were conducted in the field. The heading date for each plot was recorded when 50% of the panicles had emerged from the rice culms, as determined using visual estimation. The height and tillers of each plot were assessed at the mature stage using three central individuals, and the plant height was measured from the ground to the tip of the rice panicle (*Counce, Keisling & Mitchell,*

*2000*). The three central individuals of each plot were then harvested and air-dried in a greenhouse for biomass evaluation.

## Genotyping and genetic analysis

DNA was extracted from each RIL of the two populations and their parents following the CTAB method described by *Hulbert & Bennetzen (1991)*. The straighthead-linked markers, namely, RM282, RM225, RM2, AP3858-1, and RM206, were used to screen the RILs of the two populations.

DNA amplification was performed as previously described (*Pan et al., 2012*). As to genotyping, alleles corresponding to resistant or susceptible parents were noted as "a" or "b," respectively. RILs with both alleles were noted as "h." Missing data were noted as ".". According to our previous report, "a" was a resistant allele and "b" was susceptible at each QTL locus of the ZHE733/R312 population. In the Cocodrie/Jing185 population, "a" was notably resistant and "b" was the susceptible allele at the *qSH-8* locus, whereas "a" was the susceptible allele and "b" was the resistant allele at the *qSH-3* locus. RILs with straighthead ratings ≤ 4.0 were selected for further allelic analysis using a number of markers. These markers, including RM225, RM2, RM206, RM282, and AP3858-1, were associated with straighthead resistance (*Pan et al., 2012*) and can be useful in MAS.

## Identification of RILs and statistical analysis

In the Cocodrie/Jing185 population, RILs with over 50% Cocodrie genetic background were selected for further analysis. The agronomic traits of these selected RILs were analyzed using analysis of variance (ANOVA). Duncan's multiple range test was performed between selected RILs and Cocodrie based on the agronomic traits. RILs with different allele combinations were compared with RILs without any resistant alleles (RWARA) using the F-test and *T*-test. All these statistical procedures were conducted using SAS software v9.1 (SAS Institute Inc., Cary, NC, USA).

# RESULTS

## Gene effect of straighthead-related QTLs

Four SSR markers linked to straighthead-resistant QTLs, namely, RM225 (*qSH-6*), RM2 (*qSH-7*), RM206 (*qSH-11*), and AP3858-1 (*qSH-8*), were identified in the Zhe733/Jing185 population in a previous study (*Pan et al., 2012*). Of these QTLs, 5 RILs with different genotypes were compared based on straighthead rating, with the two parents (susceptible parent R312, which had a straighthead rating of 8.8, and resistant parent R312, which had a straighthead rating of 1.2) set as controls. The results (Fig. 2A) show that ZR-64, which had susceptible alleles at the four loci, had the highest straighthead rating (8.7). By contrast, the other four RILs (ZR-238, ZR-132, ZR-14, and ZR-83), which have at least one resistant allele each, showed lower straighthead ratings than other RILs which have none. In particular, ZR-83, which has four resistant alleles, had the lowest straighthead rating (1.3).

Two SSRs linked to the straighthead-related QTLs RM282 (*qSH-3*, susceptible QTL) and AP3858-1 (*qSH-8*, resistant QTL) were identified in the Cocodrie/Jing185 population

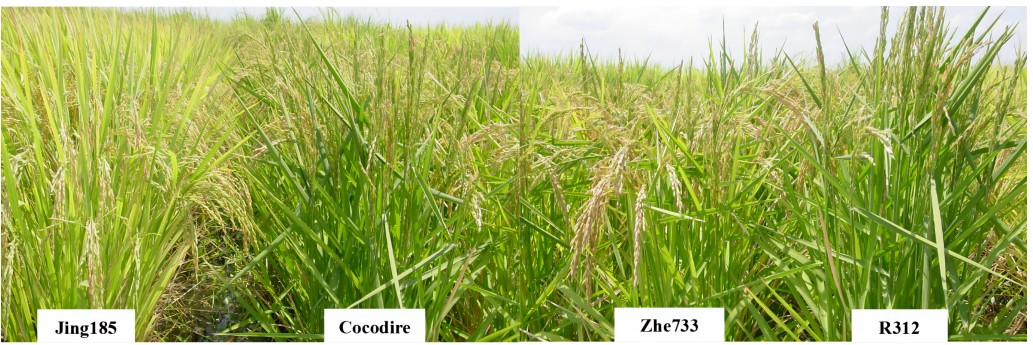

**Figure 1** Straighthead phenotype of parents, Cocodire (susceptible)/Jing185 (resistant) and Zhe733 (resistant)/R312 (susceptible).

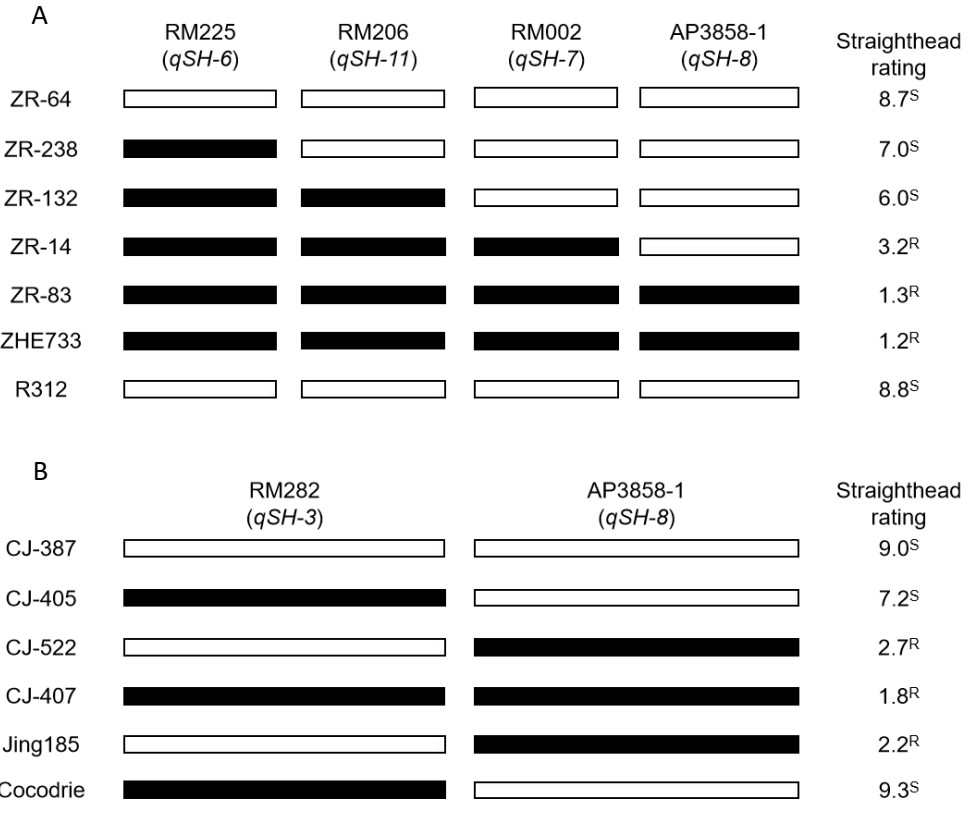

**Figure 2** Straighthead rating of RILs with different genotype in Zhe733/R312 population (A) and Cocodrie/Jing185 population (B). SSR markers RM282, RM225, RM2, AP3858-1 and RM206 were previously identified to associated with five straighthead resistant QTLs *qSH-3*, *qSH-6*, *qSH-7*, *qSH-8* and *qSH-11*, respectively. Black bar represents resistant allele, white bar represents susceptible allele in MASA-induced field. [S]: susceptible. [R]: resistant.

in a previous study (*Pan et al., 2012*). Four RILs were selected for comparison based on the straighthead ratings. The two parents (the susceptible parent "Cocodrie" with a straighthead rating of 9.3 and the resistant parent "Jing185" with a straighthead rating of 2.2) were set as controls. The results (Fig. 2B) show that RIL CJ-405, which has no resistant alleles at both loci, showed a very high straighthead rating of 9.0. CJ-522, with one resistant allele at RM282, showed a straighthead rating of 7.2. CJ-407, which has resistant alleles only at AP3858-1, showed a straighthead rating of 2.7. Furthermore, CJ-427, which has both resistant alleles, showed a straighthead rating of 1.8. Clearly, *qSH-8* showed the highest contribution to resistance. Therefore, the RILs CJ-407 and CJ-427, which have the major resistant QTL, can be used as elite lines for future straighthead-resistance breeding programs.

## Allelic analysis of straighthead-related QTLs in Zhe733/R312 and Cocodrie/Jing185 populations

To investigate the effects of the five straighthead-related QTLs, 147 RILs from Zhe733/R312 (Table S2) and 91 RILs (Table S3) from Cocodrie/R312 were used in this study. In the Zhe733/R312 population (Fig. 3A and Table 1), 16 RILs without any resistant allele (RWARA-ZR) exhibited a mean straighthead rating of 8.66. Six RILs with the resistant allele *qSH-6* (RM225) exhibited a mean straighthead rating of 8.18. Similarly, RILs with the resistant alleles *qSH-7* (RM2) and *qSH-11* (RM206) showed mean straighthead ratings of 8.55 and 8.29, respectively. Eight RILs with resistant alleles at the three loci (*qSH-6* \**qSH-7* \**qSH-11*) exhibited a much lower mean straighthead rating of 3.0. Seven RILs with the resistant allele *qSH-8* (AP3858-1) showed a mean straighthead rating of 5.24. Moreover, RILs containing combinations of *qSH-8* (AP3858-1) and any of the other three loci showed mean straighthead ratings of 5.80 (*qSH-11* \**qSH-8*), 4.88 (*qSH-6* \**qSH-8*), and 4.45 (*qSH-7* \**qSH-8*). RILs with the three resistant alleles showed mean straighthead ratings of 2.84 (*qSH-6\*qSH-7\*qSH-11*), 1.75 (*qSH-6\*qSH-7\*qSH-8*), 2.11 (*qSH-6\*qSH-11\*qSH-8*), and 1.95 (*qSH-7\*qSH-11\*qSH-8*). The lowest straighthead rating (1.64) was identified in the five RILs with resistant alleles at all four loci (*qSH-6* \**qSH-7* \**qSH-11* \**qSH-8*). Significant differences were found between all resistant RILs and RWARA-ZR, whereas no significant differences were observed between all susceptible RILs and RWARA-ZR (Fig. 3A).

In the Cocodrie/Jing185 population (Fig. 3B, Table 2), 15 RILs with no resistant allele at both loci (RWARA-CJ) exhibited the highest mean straighthead rating of 8.41. Sixteen RILs with one resistant allele, *qSH-3* (RM282), showed a mean straighthead rating of 8.07. Twenty-two RILs with only the resistant allele *qSH-8* (AP3858-1) showed a mean straighthead rating of 4.51. Eleven RILs with both *qSH-3* and *qSH-8* exhibited the lowest mean straighthead rating of 3.62. Significant differences were observed between the RILs with *qSH-8* and those with both *qSH-3* and *qSH-8* and RWARA-CJ, whereas no significant differences were found between RILs with *qSH-3* and RWARA-CJ (Fig. 3B).

## Agronomic analysis of both RIL populations and straighthead-resistant RILs

When we performed water management, we did not observe straighthead symptoms in both parents and in the 91 RILs of the Cocodrie/Jing185 population. This result shows

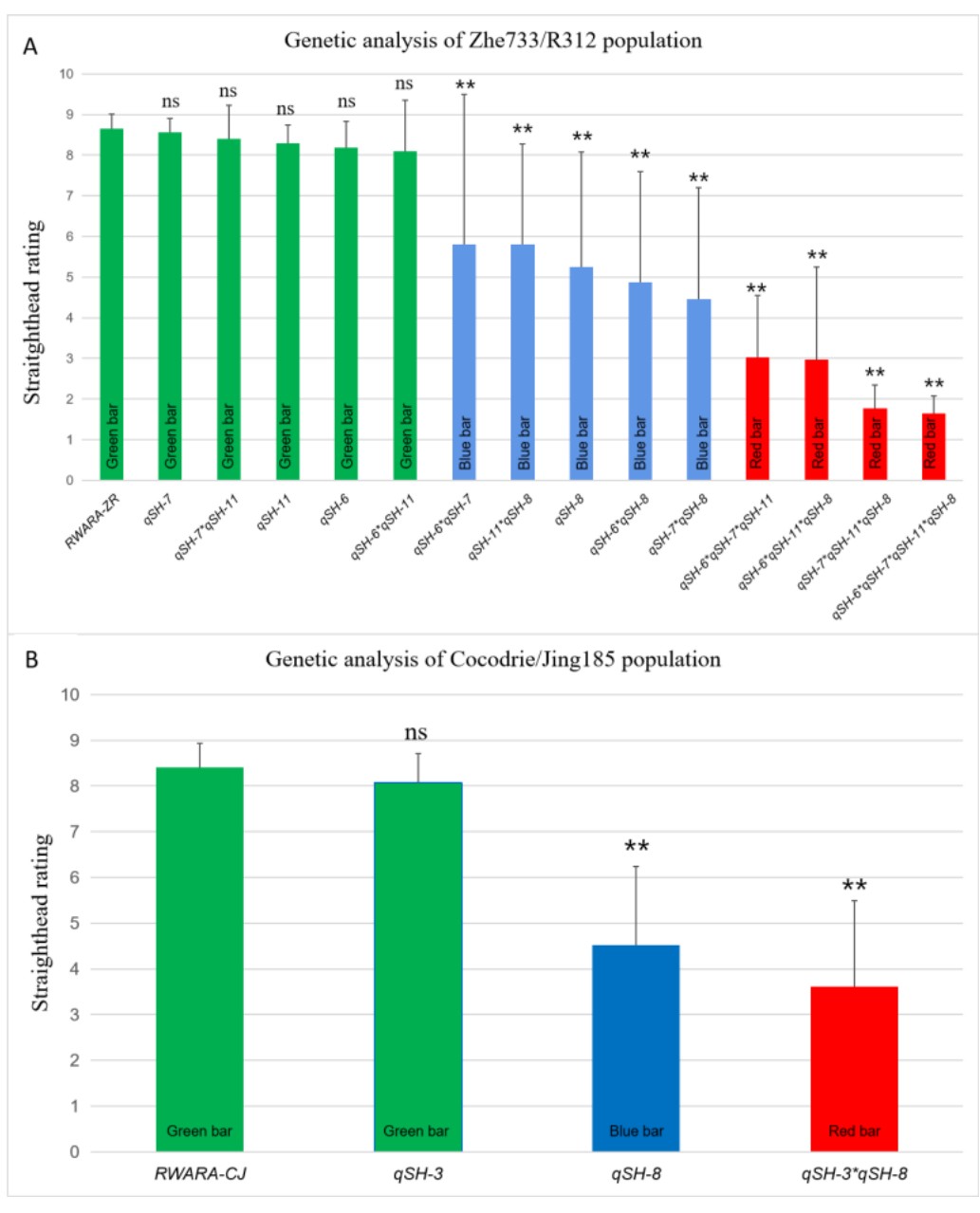

**Figure 3** **Straighthead rating of RILs with different genotype in MASA-induced Zhe733/R312 population (A) and Cocodrie/Jing185 population (B).** Green bar represents susceptible phenotype with straighthead rating above 6, blue bar represents medium phenotype with straighthead rating between 6 and 4, red bar represents resistant phenotype with straighthead rating blow 4. RWARA-ZR: RILs without any resistant allele in Zhe733/R312 population. RWARA-CJ: RILs without any resistant allele in Cocodrie/Jing185 population. ns: indicates not significant ($p > 0.05$). **: represents RILs significantly different from RWARA-ZR or RWARA-CJ at the 0.01 probability level in each population, respectively.

**Table 1  Genetic analysis of straighthead-associated QTLs in MASA-induced Zhe733/R312 population.**

| QTLs | Genotype[1] | No. of RILs | Straighthead rating[2] |
|---|---|---|---|
| RWARA-ZR | b*b*b*b | 16 | 8,66 ± 0.35 |
| qSH-6 | a | 6 | 8.18 ± 0.65 |
| qSH-7 | a | 13 | 8.55 ± 0.35 |
| qSH-11 | a | 10 | 8.29 ± 0.45 |
| qSH-8 | a | 7 | 5.24 ± 2.83 |
| qSH-6*qSH-7 | a*a | 5 | 5.80 ± 3.79 |
| qSH-6*qSH-11 | a*a | 5 | 8.10 ± 1.24 |
| qSH-7*qSH-11 | a*a | 6 | 8.40 ± 0.83 |
| qSH-6*qSH-8 | a*a | 9 | 4.88 ± 2.71 |
| qSH-7*qSH-8 | a*a | 13 | 4.45 ± 2.71 |
| qSH-11*qSH-8 | a*a | 4 | 5.80 ± 2.48 |
| qSH-6*qSH-7*qSH-11 | a*a*a | 3 | 3.02 ± 1.54 |
| qSH-6*qSH-7*qSH-8 | a*a*a | 7 | 1.55 ± 0.48 |
| qSH-6*qSH-11*qSH-8 | a*a*a | 6 | 2.96 ± 2.28 |
| qSH-7*qSH-11*qSH-8 | a*a*a | 7 | 1.95 ± 0.67 |
| qSH-6*qSH-7*qSH-11*qSH-8 | a*a*a*a | 5 | 1.64 ± 0.44 |

**Notes.**
RILs, recombinant inbred lines; RWARA-ZR, RILs without any resistant allele in Zhe733/R312 population.

[1]"a" represents resistant alleles of parent "Zhe733" and "b" represents susceptible alleles of parent "R312".

[2]Straighthead rating using a 1-9 scale. Straighthead rating of 4 or below was resistant and 6 or above was susceptible.

**Table 2  Genetic analysis of straighthead-associated QTLs in MASA-induced Cocodrie/Jing185 population.**

| QTLs | Genotype[1] | No. of RILs | Straighthead rating[2] |
|---|---|---|---|
| RWARA-CJ | b*a | 15 | 8.41 ± 0.53 |
| qSH-3 | b | 16 | 8.07 ± 0.64 |
| qSH-8 | a | 22 | 4.51 ± 1.73 |
| qSH-8 *qSH-3 | a*b | 11 | 3.62 ± 1.86 |

**Notes.**
RILs, recombinant inbred lines; RWARA-CJ, RILs without any resistant allele in Cocodrie/Jing185 population.

[1]"a" represents susceptible alleles of parent "Jing185" and "b" represents resistant alleles of parent "Cocodrie" at qSH-3 locus, meanwhile, "a" represents resistant alleles of parent "Jing185" and "b" represents susceptible alleles of parent "Cocodrie" at qSH-8 locus.

[2]Straighthead rating using a 1-9 scale. Straighthead rating of 4 or below was resistant and 6 or above was susceptible.

that straighthead was successfully prevented by the water management. The frequency distributions of the four traits, namely, heading date, plant height, tillers, and biomass, were then separately investigated (Fig. 4). The ANOVA results of the four traits show that the four traits significantly differed among the RILs from the Cocodrie/Jing185 population ($p < 0.01$) (Table 3).

A total of 27 straighthead-resistant RILs with at least one resistant allele at AP3858-1 were selected for analysis. Afterward, 166 polymorphism markers were used to compare the genetic backgrounds of the selected RILs and their susceptible parent Cocodrie. The results show that five RILs, namely, CJ-404, CJ-407, CJ-479, CJ-480, and CJ-506, shared more than 50% genotypic background with Cocodrie (Table 4), with RIL506 showing the

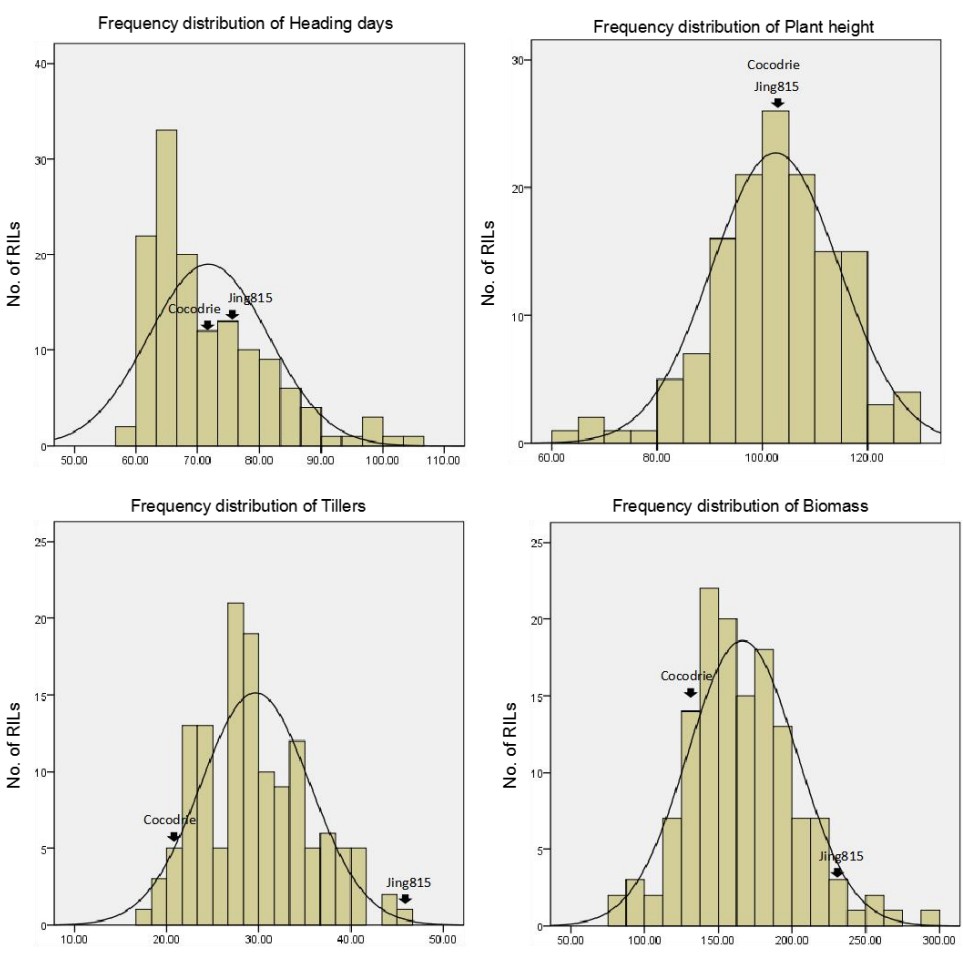

**Figure 4** **Distribution of RILs of Cocodrie/Jing185 population under water-management.**

highest genetic similarity at 66.0%. These RILs and the two parents were then subjected to phenotypical similarity analyses using Duncan's multiple test (Tables 5 and 6). Significant differences were observed between the heading days of Cocodrie and all RILs (Table 6). CJ-479 had the longest heading day among the RILs, whereas CJ-480 had the shortest one (Table 5). Significant differences in plant height were also observed between all RILs and Cocodrie, except for RIL480 (Table 6). CJ-479 had the highest plant height, whereas CJ-506 had the shortest one (Table 5). However, no significant differences were observed in the tillers and biomass of the RILs with a Cocodrie background (Table 6). In conclusion, all five RILs with greater than 50% genotypic similarity to Cocodrie showed high yields similar to Cocodrie's. These RILs are potential germplasms for straighthead-resistance breeding.

## DISCUSSION

With the discovery and application of molecular markers in the late 1970s, MAS has provided a time-saving and purpose-directing strategy for plant breeding that is superior to conventional strategy. Previous studies reported MAS application in different species

**Table 3** One-way ANOVA of four agronomic traits under water-management.

| | Source | Sum of squares | Degrees of freedom | Mean square | F-value | *P*-value |
|---|---|---|---|---|---|---|
| Heading days | Mean squared between | 27,210.667 | 92 | 295.768 | 49.148 | 4.621E−95[**] |
| | Mean squared error | 1,119.333 | 186 | 6.018 | | |
| | Total | 28,330 | 278 | | | |
| Height | Mean squared between | 39,906.708 | 92 | 433.769 | 29.036 | 1.040E−72[**] |
| | Mean squared error | 2,659.144 | 178 | 14.939 | | |
| | Total | 42,565.852 | 270 | | | |
| Tillers | Mean squared between | 15,604.872 | 92 | 169.618 | 2.913 | 6.816E−09[**] |
| | Mean squared error | 10,304.925 | 177 | 58.22 | | |
| | Total | 25,909.797 | 269 | | | |
| Biomass | Mean squared between | 3,55,919.904 | 92 | 3,868.695 | 2.743 | 1.042E−07[**] |
| | Mean squared error | 2,31,262.864 | 164 | 1,410.139 | | |
| | Total | 5,87,182.768 | 256 | | | |

**Notes.**
[**]Significantly different from zero at the 0.01 probability level.

**Table 4** Genotypic similarity analysis of RILs of MASA-induced Cocodrie/Jing185 population.

| RILs | *qSH-3* genotype[1] | *qSH-8* genotype[2] | Ancestry of cocodrie | Straighthead rating[3] |
|---|---|---|---|---|
| CJ-404 | a | a | 50.64% | 3.67 ± 1.63 |
| CJ-407 | b | a | 53.91% | 1.83 ± 0.75 |
| CJ-479 | b | a | 52.42% | 2.67 ± 1.03 |
| CJ-480 | b | a | 52.40% | 3.50 ± 1.83 |
| CJ-506 | b | a | 66.02% | 2.33 ± 1.03 |
| CJ-388 | a | a | 49.62% | 3.00 ± 1.26 |
| CJ-427 | a | a | 47.73% | 2.00 ± 1.26 |
| CJ-478 | a | a | 44.53% | 3.83 ± 1.83 |

**Notes.**
[1]"a" represents susceptible alleles of parent "Jing185" while "b" represents resistant alleles of parent "Cocodrie" at *qSH-3* locus.
[2]"a" represents resistant alleles of parent "Jing185" and "b" represents susceptible alleles of parent "Cocodrie" at *qSH-8* locus.
[3]Straighthead rating using a 1-9 scale. was averaged over 3 replications each year and 2 years for which the SD was estimated. Straighthead rating of 4 or below was resistant and 6 or above was susceptible.

**Table 5** Yield-related characteristics in Cocodrie/Jing185 population under water-management.

| RILs | Heading date | Plant height (cm) | Tillers | Biomass (kg) |
|---|---|---|---|---|
| CJ-404 | 215.33 ± 0.58 | 91.44 ± 3.37 | 22.56 ± 1.83 | 125.79 ± 17.58 |
| CJ-407 | 216.67 ± 5.51 | 88.00 ± 0.94 | 31.00 ± 3.29 | 114.24 ± 25.14 |
| CJ-479 | 236.67 ± 2.89 | 127.89 ± 2.45 | 23.89 ± 0.96 | 155.68 ± 33.06 |
| CJ-480 | 212.33 ± 1.54 | 95.78 ± 8.06 | 25.17 ± 2.59 | 134.38 ± 10.77 |
| CJ-506 | 215.00 ± 0.00 | 65.11 ± 3.01 | 29.39 ± 8.18 | 102.73 ± 11.03 |
| Cocodire | 223.00 ± 1.00 | 102.67 ± 2.91 | 19.89 ± 6.62 | 147.98 ± 41.26 |

**Table 6  Duncan's multiple tests of yield-related characteristics in Cocodrie/Jing185 population under water-management.**

| RILs | N | Duncan grouping[1] | | | |
|---|---|---|---|---|---|
| | | Heading date | Plant height | Tillers | Biomass |
| CJ-404 | 3 | B | B | A | A B |
| CJ-407 | 3 | A B | B | A | A B |
| CJ-479 | 3 | D | E A | A | A B |
| CJ-480 | 3 | A B | B C | A B | A B |
| CJ-506 | 3 | B A | A | A | |
| Cocodrie | 3 | C | C | A | A B |

**Notes.**

[1] Means with the same letter are not significantly different.

and traits (*Chen et al., 2008*; *Huang et al., 1997*; *Li et al., 2018*; *Zhao et al., 2012*). According to our previous report (*Pan et al., 2012*), the straighthead-resistant QTL *qSH-8* accounted for approximately 67% of the phenotypic variations in the Cocodrie/Jing185 population, which is much higher than those of any other QTL. In the present study, AP3858-1 tightly linked to the major straighthead-resistant QTL *qSH-8* was used to screen 91 RILs from the Cocodrie/Jing185 population. The results show that 22 RILs with the resistant allele *qSH8* (AP3858-1) showed a mean straighthead rating of 4.51 (medium resistant). This result suggests that AP3858-1 is a reliable marker for straighthead-resistance selection. The three other QTLs in the Zhe733/R312 population, namely, *qSH-6*, *qSH-7*, and *qSH-11*, accounted for 13%, 12%, and 8% of the phenotypic variations, respectively. Although the three QTLs accounted for much lower variations than *qSH-8*, they can still be useful when applied in other genetic backgrounds and can also help us understand the genetic structure of the interest trait. For instance, 49 QTLs for 14 rice traits were reported by *Wang et al. (2011)*, eight of these QTLs were related to spikelet number per panicle and to 1000-grain yield, which account for approximately 8% and 10% of the phenotypic variations, respectively. These QTLs were introduced into chromosome segment substitution lines, which exhibited increased panicle and spikelet sizes compared with their parent 93-11 (*Zong et al., 2012*). Based on our study, RILs that pyramid all three QTLs showed increased levels of straighthead resistance compared with the susceptible parent R312. This result suggests that the three QTLs can be used in MAS for resistance breeding.

In our study, the QTLs were related to MASA-induced straighthead. In previous studies on As-plant interaction, a number of QTLs were identified to correlate with As tolerance (*Ehasanullah & Meetu, 2018*; *Syed et al., 2016*; *Xu et al., 2017*) and accumulation (*Song et al., 2014*; *Wang et al., 2016*; *Yamaji & Ma, 2011*). Interestingly, some of these QTLs shared regions with our straighthead-resistant QTLs in rice. For instance, *Syed et al. (2016)* reported three QTLs, namely, *qAsTSL8*, *qAsTRL8*, and *qAsTRSB8*, which were associated with shooting length, root length, and root-shooting biomass under As stress, respectively. *Wang et al. (2016)* reported a gene, *OsPT8*, that was related to AsV transport in root cells and root-elongation inhibition. *Kuramata et al. (2013)* reported the *qDMAs6.2* gene, which was associated with As accumulation in rice grains. Thus, researchers have already connected straighthead to As accumulation. *Yan et al. (2008)* reported that the As concentration in

the straighthead-resistant cultivar Zhe733 was much lower than in the susceptible cultivar Cocodrie when the two were planted in the same soil condition. *Hua et al. (2011)* also found that the As concentration in Cocodrie was nearly three times higher than in Zhe733 when the two were grown in MASA soil. Therefore, the straighthead-resistant QTLs may also be tolerant to As stress. These QTLs will help in understanding the mechanism behind As transportation and accumulation in plants.

Although breeding for straighthead resistance has been conducted since the 1950s, little progress has been made until 2002 (*Yan et al., 2002*). One of most important factors was the lack of resistant germplasms in the US. The southern United States produces over 80% of rice, and 90% of the cultivars grown here are *tropical japonica* (*Mackill & Mckenzie, 2002*); most of these cultivars are susceptible to straighthead. In previous studies, 42 resistant accessions were identified from a survey of 1,002 germplasms collected worldwide. None of these accessions were *japonica* (*Agrama & Yan, 2010*), whereas most of the resistant accessions were classified into the *indica* subspecies. Possibly, straighthead resistance comes from *indica*. This resistance would thus be used to improve the susceptible cultivars grown in the southern U.S.. In fact, the two resistant parents in the present study are both from *indica* accessions. However, incompatibilities between the two subspecies were observed. Straighthead evaluation is based on rice infertility; therefore, the incompatibility made it challenging to obtain well-developed seeds and may have also caused bias when the straighthead resistance of the offspring was evaluated. In our previous research, for instance (*Pan et al., 2012*), 13 RILs with resistant alleles showed high straighthead ratings in some cases because of the incompatibility between the two subspecies. In the present study, we identified five $F_9$ RILs from the crossing between *japonica* Cocodrie and *indica* Jing185. These RILs had the highly straighthead-resistant QTL *qSH-8*, which is similar to Cocodrie both genotypically and phenotypically. The results suggest that the five $F_9$ RILs, which have both *japonica* genetic backgrounds and straighthead resistance, are potential lines for developing *japonica* cultivars for straighthead-resistance breeding.

## CONCLUSIONS

This study suggests that *qSH-8* is a major QTL for straighthead resistance, and AP3858-1, which is linked to *qSH-8*, is an ideal tool in marker-assisted breeding for straighthead resistance. Five RILs from the Cocodrie/Jing185 $F_9$ population contained the resistant alleles of *qSH-8*. In addition, these RILs had more than 50% genotypic background similarities to Cocodrie. Compared with Cocodrie, these lines exhibited significant differences in the heading date and plant height but no significant difference in the tillers and biomass. Most importantly, these RILs exhibited high yields similar to Cocodrie's. The genotypically and phenotypically diverse RILs are potential germplasms that can be used in straighthead-resistance breeding.

## ACKNOWLEDGEMENTS

The authors thank Hongjun Liu for the critical reviews and Tiffany Sookaserm and Yao Zhou for the technical assistance.

### Funding

The authors received no funding for this work.

### Competing Interests

The authors declare that there are no competing interests.

### Author Contributions

- Xuhao Pan conceived and designed the experiments, performed the experiments, analyzed the data, prepared figures and/or tables, authored or reviewed drafts of the article, and approved the final draft.
- Yiting Li conceived and designed the experiments, analyzed the data, prepared figures and/or tables, authored or reviewed drafts of the article, and approved the final draft.
- Xiaobai Li conceived and designed the experiments, performed the experiments, prepared figures and/or tables, authored or reviewed drafts of the article, and approved the final draft.

### Data Deposition

   The raw measurements are available in the Supplementary Files.

### Supplemental Information

Supplemental information for this article can be found online at http://dx.doi.org/10.7717/peerj.14866#supplemental-information.

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
