# Peer review of "Quantitative trait loci associated with straighthead-resistance used for marker assisted selection in rice (Oryza sativa L.) RIL populations"

_PeerJ, doi:10.7717/peerj.14866_

## Round 0.1 · original submission · Minor Revisions

Dear Authors

Please revise the MS as per comments of the reviewers and resubmit the MS

Please check the language throughout the MS by using fluent English speakers

Reviewer 1 ·

Basic reporting

The barplots should include statistical significances

Experimental design

no comment

Validity of the findings

no comment

Additional comments

In this article, Pat et al. has studied five known QTLs of Straighthead disorder of rice plants in two novel recombinant inbred lines of Jing185/Cocodrie and Zhe733/R312. Using the F9 populations of these breads they then obtained the genotypes corresponding to the QTL loci and characterized their effects on Straighthead phenotype.
The reviewer thinks that for its scope, the study was well conducted and describes process reasonably well. Apart from thorough check for the grammatical mistakes, there are a few points to be made:
- The standard deviations of the bars depicted in the Figure 3 are too wide for the medium phenotypes. Is there a way to get more data for these QTLs? If not, the blue bars hardly can represent a signal.
- Conversely, the statistical significances between the distributions should be indicated at least.
- It is not very clear on what basis the “straigthead rating” values were classified to susceptible, medium and resistant phenotypes. An ad-hoc threshold for defining the boundaries is fine, but it should be discussed and at least mentioned if it is purely an ad-hoc boundary or is it based on previous literature.

Reviewer 2 ·

Basic reporting

No comment

Experimental design

No comment.

Validity of the findings

No comment

Additional comments

Comments:
Straighthead disease is a complex conventional physiological disorder of rice and affects rice cultivation worldwide. Pyramiding QTLs using marker-assisted selection (MAS) is effective for improving straighthead-resistance in rice breeding programs. Authors have done a lot of works including exploration of QTL effects and selection of elite lines with straighthead-resistance. These markers linked to straighthead-resistant QTL will be very helpful in QTL pyramiding for improving of straighthead-resistance in rice. However, there are some minor errors listed as below. The language need to be polished.

Minor errors:
1. There are some grammatical and/or spelling mistakes in manuscript. Please overview the manuscript for more details.
2. Line 1: “maker assistant selection” should be corrected as “maker assisted selection”;
3. Line 147: “Straihthead” should be “straighthead”;
4. Line 178: “qSH-3” and “qSH-8” should be italicized;
5. Line 212: “link” should be “linked”.
6. Line 167: What’s the interval of the fine-mapped QTL qSH-8? It was not mentioned in manuscript.
7. Line 215-217, The accumulated contribution (33%) of three QTLs (qSH-6, qSH-7, and qSH-11) were lower than qSH-8 (42%), however the RILs having all the three resistant QTL alleles showed lower straighthead rating of 3.0 than that (5.24) of the RILs having qSH-8 alleles, please explain why?
8. In addition, the reference style in this manuscript is not completely correct, and needs to be verified carefully.

---

## Round 0.2 · Minor Revisions

The manuscript is almost ready but please see the following comment from a Section Editor:

> The manuscript may actually be fine for scientific content; however, at this stage it is difficult to read with a detected language barrier. I recommend that this manuscript be proofed before further evaluation. A partial markup of the manuscript is included.

Reviewer 1 ·

Basic reporting

no comment in addition to the initial version.

Experimental design

no comment in addition to the initial version.

Validity of the findings

no comment in addition to the initial version.

Additional comments

The authors have addressed all the questions and remarks of the reviewer and the reviewer thinks that the manuscript can be published.

Reviewer 2 ·

Basic reporting

No comments

Experimental design

No comments

Validity of the findings

No comnments

---

## Round 0.3 · Minor Revisions

Dear Authors
The language must be checked by a proficient English speaker or use editing services and resubmit for consideration.

Please provide an editing certificate or a note from the person who does the editing.

---

## Round 0.4 · accepted · Accept

Now the manuscript is in acceptable form